# COVID-19 Vaccination in Inflammatory Bowel Disease (IBD)

**DOI:** 10.3390/jcm11092676

**Published:** 2022-05-09

**Authors:** Aleksandra Kubas, Ewa Malecka-Wojciesko

**Affiliations:** Department of Digestive Tract Diseases, Medical University of Lodz, Kopcinskiego 22, 90-153 Lodz, Poland; aleksandra.kubas@stud.umed.lodz.pl

**Keywords:** inflammatory bowel disease, COVID-19 vaccination, biologic therapy, vaccine safety, vaccine effectiveness, vaccination willingness

## Abstract

Vaccines against SARS-CoV-2 are believed to play a key role in the suppression of the COVID-19 pandemic. However, patients suffering from inflammatory bowel diseases (IBD) were excluded from SARS-CoV-2 vaccines trials. Therefore, concerns regarding vaccination efficacy and safety among those patients were raised. Overall, vaccination is well tolerated in the IBD population, and different gastroenterological societies recommend vaccinating patients with IBD at the earliest opportunity to do so. Nevertheless, very little is known about the safety of COVID-19 vaccines in special IBD populations such as pregnant and breastfeeding women or pediatric patients, and further research on this matter is crucial. The available data on vaccine efficacy are promising and show high seroconversion rates in IBD patients on different immune-modifying therapies. However, patients treated with high doses of systemic corticosteroids, infliximab or infliximab and immunomodulators may have a blunted response to the vaccination. The data on COVID-19 vaccination willingness among patients with IBD are conflicting. Nevertheless, vaccine effectiveness and safety are reported to be the most common reasons for hesitancy. This review examines the effectiveness and safety of COVID-19 vaccines and describes vaccination willingness and the reasons for potential hesitancy among patients with IBD.

## 1. Introduction

Following its initial discovery in Wuhan in December 2019, the novel severe acute respiratory syndrome coronavirus-2 (SARS-CoV-2) soon became the agent behind the global COVID-19 pandemic. Due to its high morbidity and mortality rates, preventing COVID-19 became the highest priority for public health and led to the unprecedentedly quick development and authorization of vaccines against SARS-CoV-2. 

The inflammatory bowel diseases (IBD), comprising Crohn’s disease (CD) and ulcerative colitis (UC), are characterized by chronic inflammation of the gastrointestinal tract resulting in progressive bowel damage and, subsequently, progressive dysfunction. Therefore, the main aim in treating IBD patients is to control the inflammation, often by the administration of various immunosuppressive drugs [1]. Such therapy puts patients with IBD at an increased risk of infections [2]. However, according to the current evidence, IBD itself is not a risk factor for the acquisition or higher severity of SARS-CoV-2 infection [3,4,5,6,7]. Nevertheless, at the beginning of the COVID-19 pandemic, the British Society of Gastroenterology (BSG) COVID-19 IBD Working Group classified IBD patients treated with ≥20 mg prednisolone or equivalent per day, or who started biologic plus immunomodulator or systemic steroid therapy within the previous six weeks, to be at a high risk of incurring severe complications from COVID-19 [8]. However, in a more recent study, Brenner and Ungaro et al. reported older age, ≥2 comorbidities and the use of systemic corticosteroids and sulfasalazine or 5-aminosalicylate to be the risk factors for severe COVID-19 in the IBD population [9]. 

Currently, vaccination against SARS-CoV-2 is believed to play a key role in the management of the COVID-19 pandemic and in the prevention of severe disease. Data on the efficacy and safety of the vaccines are promising in the general population [10]. However, as neither IBD patients nor any patients managed with immunosuppressive therapies were included in the phase III-trials for COVID-19 vaccines, vaccination against COVID-19 in this patient group is associated with two main concerns: firstly, that vaccine efficacy may be blunted by immunomodulators, biologics and corticosteroids, and secondly, that vaccination may cause the exacerbation of an underlying inflammatory disease [11]. The emerging data considering vaccination effectiveness in IBD patients are inconsistent and often limited by a short follow-up period. Moreover, very little is known about the safety of COVID-19 vaccines in special IBD populations such as pregnant and breastfeeding women or pediatric patients. 

## 2. Available SARS-CoV-2 Vaccines

At the time of writing, five COVID-19 vaccines are authorized for use in the European Union by the European Medicines Agency (EMA). Two mRNA vaccines (Pfizer-BioNTech BNT162b2 and Moderna mRNA-1273), two non-replicating viral vector vaccines (AstraZeneca and COVID-19 Vaccine Janssen) and one recombinant protein vaccine (Novavax). mRNA vaccines contain an RNA molecule of a target antigen encapsulated in lipid nanoparticles. After the injection, the mRNA-lipid nanoparticle complex is incorporated into the host cells and translated into the SARS-CoV-2 spike protein. This protein is a strong antigen that stimulates both the humoral and cellular immune responses of the vaccinated person. Viral vector vaccines comprise genetically modified viruses with genetic information of the foreign antigen incorporated into their genome. After the vaccination, viral vectors use host translational machinery in order to express the SARS-CoV-2 spike protein. The protein subunit platform acts by expressing the SARS-CoV-2 S protein or only a part of it, such as the receptor-binding domain (RBD), in various expression systems (for example, insect cells or yeast cells) [12,13]. Currently, only two vaccines, Pfizer-BioNTech and Moderna, are registered for use in the pediatric population, i.e., for children aged over 5 years and 12 years, respectively. However, on 24 February 2022, the EMA recommended granting an extension of indication for the Moderna vaccine in children aged 6 to 11 [14]. The vaccination schedule of the COVID-19 vaccines available at the time of writing is presented in Table 1.

According to the current recommendations by the EMA, all people over the age of 12 should receive a booster dose of the COVID-19 vaccine, preferably the mRNA vaccine [15]. The booster dose is an injection given after the completion of the primary series of the vaccination, this being the second dose of the COVID-19 Vaccine Janssen and the third dose for all other vaccines. Immunocompromised individuals, including IBD patients treated with high doses of corticosteroids (≥20 mg prednisone or equivalent per day) or with any other immunosuppressive drugs, are at risk of a blunted response to the initial vaccine series. Therefore, it is recommended that moderately to severely immunocompromised patients get an additional dose of the COVID-19 vaccine at least 28 days after the final injection of the primary series [15]. At the time of writing, only two studies have investigated the immune response to the third dose of the COVID-19 vaccine among IBD patients. Both studies reported a boost in the seroconversion rate after an additional dose. Schell et al. report that 97.1% of tested IBD patients had detectable antibody concentration after the initial vaccination series, and this value rose to 100% after administration of the additional dose. Importantly, two patients who did not seroconvert after the primary series were seropositive after the third dose of the vaccine [16]. Long et al. also report an improvement in the seroconversion rate after an additional dose of the SARS-CoV-2 vaccine among patients with IBD, with 93% of patients seroconverting after the initial vaccination series and 99.5% after additional immunization. Furthermore, 45 of 47 (95.7%) individuals who failed to produce detectable antibody levels after the initial series seroconverted after the additional dose of the COVID-19 vaccine [17].

## 3. SARS-CoV-2 Vaccine Efficacy in Patients with IBD

Data on effectiveness of available SARS-CoV-2 vaccinations in IBD patients are limited but emerging. There are two main concerns regarding COVID-19 vaccine efficacy: whether patients with IBD are able to mount a sufficient, long-lasting immune response to the vaccination, and whether the immunosuppressive drugs used in IBD therapy influence the effectiveness of vaccines against SARS-CoV-2. Before the COVID-19 era, various studies found that patients with IBD who receive immunomodulators, biologics or corticosteroids may have a blunted response to certain vaccines. It was observed that individuals treated with infliximab or adalimumab had lower seroconversion rates than healthy controls after being administered with the HBV, influenza and pneumococcal vaccines [18,19,20]. On the other hand, vedolizumab did not impair the effectiveness of the HBV or influenza vaccines but was associated with a blunted response to the cholera vaccine administered orally [21]. Therefore, at the beginning of the COVID-19 pandemic, many questions have been raised about whether IBD patients receiving immune-modifying therapies are able to mount a sufficient immune response to COVID-19 vaccines. 

CLARITY-IBD is a multicenter study enrolling 6935 patients across the UK and investigating whether drugs used in inflammatory bowel disease such as infliximab, vedolizumab and/or concomitant immunomodulators (thiopurines or methotrexate) impair immunity following COVID-19 vaccination [14]. The vaccines used in this study are the Pfizer-BioNTech and AstraZeneca versions. The results indicate that after the second dose of the vaccine, individuals who have not had COVID-19 and were treated with infliximab had lower antibody levels than those treated with vedolizumab. Moreover, antibody titer declined quicker in patients who received infliximab [22,23]. Interestingly, antibody levels remained high for much longer in individuals who had COVID-19 as well as two doses of the vaccine. This stems from the fact that prior COVID-19 infection acts as another immunizing event. The CLARITY study results also indicate that after the third primary dose of the vaccine, antibody levels rose notably in both infliximab- and vedolizumab-treated patients. It is also worth mentioning that, irrespective of the immunosuppressive drug used, the recipients of the Pfizer-BioNTech vaccine had the highest antibody responses [24]. 

Researchers at the Cedars-Sinai Medical Center have been conducting the CORALE-IBD study into the safety and effectiveness of COVID-19 vaccination in IBD patients. It assessed the seroconversion rate in 582 patients with IBD who were vaccinated with mRNA vaccines and had no prior COVID-19 infection. The available outcomes were very promising, as 99% of the participants, irrespective of the immunosuppressive medications used, demonstrated seroconversion, with the peak response two weeks after the completion of the primary series of the vaccination [25]. These findings regarding seroconversion in different IBD medication groups support those of previous studies. 

The International Study of COVID-19 Antibody Response Under Sustained Immune Suppression in IBD (ICARUS-IBD), an international cohort study, has measured the serologic response in 48 IBD patients treated with different biologics and vaccinated with the Pfizer-BioNTech or Moderna vaccine. In total, 100% of patients seroconverted after two doses of vaccination, irrespective of the biologics used. Moreover, patients with prior COVID-19 infection had higher antibody levels after only one dose, which supports the belief that SARS-CoV-2 infection acts as another immunizing event [26]. The study conducted by Kappelman MD at el. on 317 IBD patients, of whom 95% had a detectable antibody response, also found that different immune-modifying therapies used in IBD do not vastly reduce the response to the mRNA COVID-19 vaccines [27]. These findings are further reinforced by data regarding the serological response to two doses of either the Pfizer-BioNTech or AstraZeneca vaccine in 126 patients with IBD receiving different biologics. A total of 74% of the patients who were on infliximab, 81% of those who were on adalimumab and almost 93% of those who were on vedolizumab, along with all of the patients who were on ustekinumab, seroconverted, irrespective of the vaccine type used [28]. Moreover, a recently published meta-analysis from Jena et al. which included 9447 patients with IBD showed that seroconversion rates were statistically similar in spite of the different drugs used in IBD therapy [29]. 

All in all, most of the available studies indicate that patients with IBD will seroconvert after SARS-CoV-2 vaccination. However, it is important to identify the predictors of possible non-seroconversion. Recently published data from the PREVENT-COVID prospective observational study indicates that the use of combination therapy with anti-TNF and immunomodulators may contribute to reduced antibody response in patients with IBD, as may the variables of a longer time since vaccination, vaccine type and older age. Even though 96% of the PREVENT-COVID participants seroconverted, a higher proportion of mRNA vaccine recipients had detectable antibody levels than the adenovirus vector vaccine recipients (96% vs. 81%) [30]. The noticeably lower antibody levels observed in patients treated with infliximab or infliximab and immunomodulators compared to those on different biologics are consistent with the results from other IBD studies [23]. 

Patients treated with infliximab demonstrated a particularly striking attenuation of immunogenicity in a UK multicenter VIP study, where immunosuppressed patients with IBD had a 10-fold reduction in antibody levels compared to healthy controls. The VIP study was also one of a few studies that assessed the response to the two doses of the COVID-19 vaccine among IBD patients treated with JAK-inhibitors. Researchers identified significantly lower anti-SARS-CoV-2 spike protein antibody concentrations in tofacitinib-treated patients in comparison to healthy controls (geometric mean concentration 430 U/mL vs. 1578 U/mL, respectively). Nevertheless, 90% of participants on infliximab monotherapy, 87% on thiopurine plus infliximab combination therapy and all on tofacitinib therapy produced a protective level of antibodies [31]. 

However, the question was raised whether an association exists between antibody titers and troughs of anti-TNFs. In a prospective study conducted by Cerna et al., the relationship between anti-SARS-CoV-2 IgG concentrations and serum trough drug levels was evaluated in 292 IBD patients treated with infliximab or adalimumab. The researchers did not observe any correlation between drug levels and serological responses to the COVID-19 vaccine [32]. Nevertheless, it is important to remember that a blunted response does not translate into vaccine inefficacy. 

Many of the conducted studies did not thoroughly characterize the correlation between antibody titer and protection from the subsequent SARS-CoV-2 infection. The most significant clinical measure for assessing vaccine effectiveness is whether it prevents disease acquisition or severe course and whether it reduces all-cause mortality. Reassuringly, in a study performed by Khan and Mahmud, patients who were fully vaccinated had a significant likelihood of remaining infection-free compared to those who were unvaccinated or only partially vaccinated. Moreover, fully vaccinated patients had a greater likelihood of being free of severe infection. Finally and most importantly, the mortality rate was significantly lower among fully vaccinated patients than unvaccinated ones [33]. 

Taken together, the emerging data provide reassurance that the various tested biologic medications do not markedly reduce the response to COVID-19 vaccines, and they support the recommendations from different gastroenterological societies that all patients with IBD should be vaccinated, regardless of immune-modifying therapies [34,35]. It has been advised that IBD patients should get two doses of the vaccination even if they have recovered from COVID-19, as data on the duration of protective antibody levels are still under examination [36]. Moreover, according to EMA recommendations, patients treated with combined immunosuppression (anti-TNF agents plus an immunomodulator) and those on corticosteroids should receive a third dose after the initial series of vaccination. Furthermore, all individuals over the age of 12 should also get a booster dose of the SARS-CoV-2 vaccine [15]. In addition, high doses of systemic corticosteroids may reduce vaccine effectiveness [30]. Therefore, the position statement on SARS-CoV2 vaccination by the British Society of Gastroenterology Inflammatory Bowel Disease section and the IBD Clinical Research Group recommends that SARS-CoV2 vaccination should be administered when patients are taking the lowest dose of systemic corticosteroid [36]. However, patients should be informed that vaccine efficacy may be blunted when receiving systemic corticosteroids. 

## 4. Safety of SARS-CoV-2 Vaccinations in Patients with IBD

Currently, the available information regarding the safety of COVID-19 vaccines in patients with IBD is limited. However, it is known that inactivated vaccinations are considered to be safe and recommended in IBD, whereas live vaccinations are contraindicated in immunosuppressed patients [34,37,38]. COVD-19 vaccines appear to be safe and well tolerated among patients with IBD, specifically mRNA vaccines such as the Pfizer-BioNTech and Moderna COVID-19 vaccines. The frequency and nature of adverse events after vaccination are similar to those in the general population [39,40,41]. However, a comparison of post-vaccination symptoms between healthcare workers without IBD and adult IBD patients found the adverse events to be less common in the IBD group, who had less severe symptoms. Only the gastrointestinal symptoms occurred more often in the IBD group compared to controls, and only after the first dose [42]. In a study conducted by Weaver KN et al. on a cohort comprising 3316 vaccinated IBD patients, the most common localized adverse event was injection site tenderness, with a prevalence of 68% after the first and second dose. Fatigue, which was the most common systemic adverse event, occurred in 46% patients after the first dose and 68% after the second [39]. Other commonly reported adverse events included pain in the injection site, malaise, fever and chills [39,41]. Studies also indicate that patients with IBD are more likely to demonstrate more severe adverse reactions after dose two than dose one. Interestingly, Botwin et al. report that adverse events occurred more frequently compared to other IBD patients among those who were less than 50 years of age and had prior SARS-CoV-2 infection. Subjects with ulcerative colitis had vaccination side effects more often than those with Crohn’s disease (78% vs. 55% after the second dose) [41]. However, apart from age <50 years, other available studies failed to find similar correlations between the mentioned factors and the more frequent occurrence of adverse events after vaccination. 

Another important concern regarding COVID-19 vaccine safety is whether it could exacerbate underlying inflammatory disease. Reassuringly, the current evidence indicates a similar exacerbation rate in vaccinated and unvaccinated patients, indicating that vaccination is not associated with the exacerbation of IBD [39,43]. 

A comparison of 707 vaccinated and unvaccinated IBD patients found no significant difference in disease outcomes between the two groups, i.e., the two had similar risks of exacerbation during the follow-up period, with adverse event incidence rates of 29% and 26%, respectively. In this case, exacerbation was defined as hospitalization, treatment escalation and the introduction of corticosteroid or topical drugs. Moreover, researchers included the number of exacerbations during the previous two years and the time interval from the last exacerbation as other matching criteria between the compared groups. [43].

The British Society of Gastroenterology (BSG), European Crohn’s and Colitis Organization (ECCO) and the International Organization for the Study of Inflammatory Bowel Diseases (IOIBD) recommend that all IBD patients should get vaccinated at the earliest opportunity, irrespective of the vaccine type offered to them, although mRNA vaccines are preferable [35,44]. Other SARS-CoV-2 vaccines, including replication-incompetent vector vaccines, inactivated vaccines and recombinant vaccines, are also considered to be safe for IBD patients [34]. It is important to remember that the recommendations given by different gastroenterological societies to take any vaccine type at the earliest opportunity were often made at the time of vaccination scarcity. Currently, the only contraindications to COVID-19 vaccination are previous severe allergic reaction or anaphylaxis to the vaccine or to any of its components [36].

Even though more and more is known about the safety of COVID-19 vaccines in IBD patients, further studies clarifying the remaining uncertainties and enrolling patients into observational registries would help to address those unanswered questions. Currently ongoing COVID-19 vaccination trials among IBD patients are presented in Table 2.

## 5. SARS-CoV-2 Vaccination in Special IBD Populations 

### 5.1. Pregnancy and Breastfeeding 

Currently, the impact of COVID-19 on pregnant and breastfeeding women with IBD is not known. According to recently published guidelines by the Royal College of Obstetricians and Gynecologists (RCOG), in the general population, pregnant women do not have an increased risk of contracting SARS-CoV-2; however, they do have a higher risk of severe disease course compared with non-pregnant women. Moreover, maternal COVID-19 infection is associated with a more frequent prevalence of adverse neonatal outcomes. Therefore, it is advisable that all pregnant women be vaccinated, regardless of pregnancy stage, preferably with the Pfizer-BioNTech or Moderna vaccines. There is also no need to stop breastfeeding in order to receive a COVID-19 vaccine [45]. 

There are no available clinical data on the efficacy and safety of SARS-CoV-2 vaccination in pregnant and breastfeeding women with IBD. Nevertheless, gastroenterological societies such as IOIBD, BSG or The Korean Association for the Study of Intestinal Diseases (KASID) recommend that pregnant or breastfeeding women with IBD should be vaccinated against SARS-CoV-2, just as pregnant or breastfeeding women in the general population should, in accordance with the government vaccination recommendations [8,34,35]. These guidelines are based on the fact that COVID-19 vaccines are considered to be safe for pregnant women without IBD, and no pregnancy-specific adverse events were registered in pregnant women vaccinated against SARS-CoV-2 [45].

### 5.2. Paediatric

Only two COVID-19 vaccines are currently available for patients under 18 years old: the Pfizer-BioNTech and Moderna vaccines. Although the Pfizer-BioNTech vaccine is not currently authorized for children below 5 years of age, there are ongoing clinical trials on its use among children aged 6 months to 4 years [46]. 

Currently, very little data exists on the safety and efficacy of COVID-19 vaccines in children with IBD. Joelynn Dailey et al. conducted a cohort study on 436 IBD patients (age range from 2 to 26 years old), which found that pediatric patients with IBD and on biologic drugs had a higher level of antibody response after vaccination compared with patients who had only SARS-CoV-2 infection [47]. These findings reassuringly indicate that vaccination is effective in children with IBD, even among those on immunosuppressive therapy. However, further research is crucial to evaluate the durability of the response to the vaccination as well as its safety profile.

Despite the limited data, it is currently recommended that all children ≥5 years old, including those with IBD, should be vaccinated against SARS-CoV-2 [48,49]. Moreover, the Centers for Disease Control and Prevention (CDC) recommends that children ages 5 through 11 years who are moderately or severely immunocompromised should receive a primary series of three doses of the Pfizer-BioNTech COVID-19 vaccine, whereas those 12 years or older should get a total of four doses [50]. 

## 6. COVID-19 Vaccination Willingness and Hesitancy in Patients with IBD

The British Society of Gastroenterology has emphasized that it is necessary to track the attitude of IBD patients towards COVID-19 vaccination and to identify any reasons for potential hesitancy about vaccination uptake [36]. Although gastroenterological societies such as The International Organization for the Study of Inflammatory Bowel Disease recommend vaccinating all patients with IBD at the earliest opportunity, some patients still remain skeptical [34,44]. The most common concerns given by patients are the efficacy and long-term safety of vaccines [51,52,53]. Reassuringly, the majority of currently available survey studies have reported a high willingness among patients with IBD to get vaccinated against SARS-CoV-2 [54]. A study conducted by Dalal et al. on patients with IBD in the United States indicated an 80.9% willingness to receive COVID-19 vaccination, which was higher than the rate among the US general population in December 2020, estimated by the Kaiser Family Foundation (71%) [55,56]. Similarly, another survey conducted by Costantino et al. found an 80.3% willingness to get vaccinated against SARS-CoV-2 in an Italian population with IBD [51]. In both mentioned studies, the authors found Caucasian type, being educated to bachelor’s level and adherence to previous vaccinations to be the most significant factors determining willingness. Moreover, several surveys found male sex to be one of the determinants of a positive attitude towards COVID-19 vaccination [51,57,58]. 

In contrast, Wu et al. found that only 16.0% of surveyed IBD patients opted for vaccination against SARS-CoV-2, while 50.7% were indecisive [52]. These findings are in accordance with another study from five German specialized IBD centers where patients suffering from IBD were more hesitant to get vaccinated compared with the control group without IBD. Only 58.5% of IBD patients planned to get a COVID-19 vaccination or had already been vaccinated, compared to 65.1% of the healthy controls [53]. However, it is worth mentioning that, regardless of the rate of vaccination intent, patients from different studies reported similar reasons for vaccination hesitancy, with the most common causes being vaccine safety and efficacy. The discrepancy with findings from different studies may stem from different cultural backgrounds and the times when the studies were performed, as well as the rapidly changing knowledge about COVID-19 vaccines and the impact of the infection itself on patients with IBD. However, it is also possible that the high rates of recorded COVID-19 vaccination willingness were overestimated, as most surveyed patients identified as Caucasian with a relatively high socioeconomic status. This is even more likely when considering the findings published by the CDC and the UK Office for National Statistics, which indicate that, in the general population, the highest vaccination refusal is noted among black persons and adults with lower educational attainment and lower income [59,60]. Therefore, further studies in more diverse patient populations with IBD are essential. 

Clinicians play a key role in achieving high COVID-19 vaccine intent among IBD patients. Therefore, it is recommended that these clinicians educate patients, identify their concerns and dispel any misconceptions about COVID-19 vaccines in the light of the increasing knowledge about the safety and efficacy of vaccines [61]. 

## 7. Conclusions

In conclusion, the existing evidence shows that vaccination against SARS-CoV-2 is highly effective in patients with IBD. The available data indicate that vaccination is not associated with a significant risk of exacerbation in most patients. In addition, the IBD population demonstrates the same or even a lower prevalence of adverse events than the general population. Taking all that into consideration, all patients with IBD should receive the primary series and a booster dose of the COVID-19 vaccination, and moderately and severely immunocompromised individuals should receive an additional dose after the primary series. Furthermore, clinicians must focus on providing patients with sound, unbiased advice in order to support their decision to vaccinate. 

## Figures and Tables

**Table 1 jcm-11-02676-t001:** COVID-19 Vaccination Schedule and Use.

Vaccine	Vaccine Type	Age Recommendations	Primary Series	Fully Vaccinated
Pfizer-BioNTech	mRNA	>5 years old	2 doses given 21 days apart	2 weeks after final dose in primary series
Moderna	mRNA	>12 years old	2 doses given 28 days apart	2 weeks after final dose in primary series
AstraZeneca	viral vector vaccine	>18 years old	2 doses given between 28 to 84 days apart	2 weeks after final dose in primary series
COVID-19 Vaccine Janssen	viral vector vaccine	>18 years old	1 dose	2 weeks after the vaccination
Novavax	recombinant protein vaccine	>18 years old	2 doses given 21 days apart	2 weeks after final dose in primary series

**Table 2 jcm-11-02676-t002:** Currently ongoing COVID-19 vaccination trials according to clinicaltrials.gov (accessed on 18 April 2022).

Study Number	Number of Cohorts	Inflammatory Bowel Disease Patients Characteristics	Healthy Controls	Non-IBD Patients	Recruitment Status as of 18 April 2022	Primary Outcomes
*NCT04818892*	2	Treated with non-systemic immunosuppressive medications or treated with systemic immunosuppressive medications	No	No	Active, not recruiting	Increase in Geometric Mean Titers of SARS-CoV-2 antibody concentrations after mRNA COVID-19 vaccination(after at least 2 doses); Sustained antibody concentrations of mRNA COVID-19 vaccines; Increase in the level of T-cell response after mRNA COVID-19 vaccine; Proportion of patients with detectable levels of T-cell response after mRNA COVID-19 vaccine
*NCT05067959*	2	Treated with anti-TNF medications or treated with medications different than anti-TNFs	Yes	No	Active, not recruiting	Percentage of patients with positive seroconversion
*NCT05014555*	4	Treated with mesalamine or thiopurine or corticosteroids, or treated with anti-TNF therapy or combination therapy of anti-TNF therapy along with either methotrexate (15 mg) or azathiprine (≥1.0 mg/kg or 6 MP 0.5 mg/kg), or treated with ustekinumab (monotherapy or combination therapy with methotrexate or azathioprine), or treated with vedolizumab (monotherapy or combination therapy with methotrexate or azathioprine)	No	No	Not yet recruiting	Increase in Geometric Mean Titers of SARS-CoV-2 antibody concentrations after COVID-19 vaccination
*NCT04769258*	2	Treated with immunomodulatory medications or not treated with immunomodulatory medications	No	No	Not yet recruiting	Quantitative serum titer of COVID-19 antibodies (IgM and IgG)
*NCT05153850*	1	Treated with biological medication and with complete vaccination schedule against SARS-CoV-2	No	No	Not yet recruiting	Seroconversion rate measured by assessing the presence of antibodies in the peripheral blood four months after completion of the COVID-19 vaccination schedule
*NCT04798625*	7	Treated with immunosuppressive medications	No	Yes	Active, not recruiting	Seroconversion (change from baseline in the serum concentration level of SARS-CoV-2 antibodies)

## Data Availability

Not applicable.

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
