# Peer review of "COVID-19 Vaccination in Inflammatory Bowel Disease (IBD)"

_jcm, 2022, doi:10.3390/jcm11092676_

Round 1

Reviewer 1 Report

This is an interesting article describing the urgent need of COVID-19 vaccination to IBD patients and common vaccine hesitancy in IBD patients. Authors should further address the following points to make this manuscript more interesting:

  1. Line 51-52: any reported study?
  2. Section 2, a detailed description of different available vaccine platforms should be referred here, authors should consider referring this article (PMID 34356617).
  3. Line 242-243: A short description in a tabular form regarding ongoing COVID-19 vaccination trials in IBD patients would be helpful. (https://www.clinicaltrials.gov/ct2/results?cond=IBD&term=covid-19&cntry=&state=&city=&dist=&Search=Search)

Author Response

Response to Reviewer 1 Comments

Point 1: Line 51-52: any reported study?

Response: We are not entirely sure which part of the lines 51-52 the reviewer is referring to. As for the exacerbation - one of the most common misconceptions among IBD patients concerning vaccination safety and efficacy reported by Malhi et al. was exacerbation of the disease. ( Gurtej Malhi, Amir Rumman, Reka Thanabalan, Kenneth Croitoru, Mark S. Silverberg, A. Hillary Steinhart, Geoffrey C. Nguyen, Vaccination in Inflammatory Bowel Disease Patients: Attitudes, Knowledge, and Uptake, Journal of Crohn's and Colitis, Volume 9, Issue 6, June 2015, Pages 439–444, https://doi.org/10.1093/ecco-jcc/jjv064 ). Mentioned citation has been added.  However, current evidence show that vaccination is not associated with the exacerbation of the underlying disease. This issue in terms of COVID-19 vaccines is addressed in a fourth paragraph with citation and discussion of studies from Weaver et al. and Lev-Tzion et al.. When it comes to Emerging data considering vaccination effectiveness in IBD patients are inconsistent and often limited by short follow-up period. mentioned studies are discussed throughout the whole review.

Point 2: Section 2, a detailed description of different available vaccine platforms should be referred here, authors should consider referring this article (PMID 34356617).

Response: A detailed description of different available vaccine platforms has been included. 

Point 3: Line 242-243: A short description in a tabular form regarding ongoing COVID-19 vaccination trials in IBD patients would be helpful. (https://www.clinicaltrials.gov/ct2/results?cond=IBD&term=covid-19&cntry=&state=&city=&dist=&Search=Search)

Response: The proper table summarizing ongoing COVID-19 vaccination trials has been included.

Reviewer 2 Report

The article is well written and would be very interesting for the journal readers. If possible, I suggest to add two references in the introduction, both of them talk about COVID and IBD (PMID: 33776370 and 33859104).

Author Response

Response to Reviewer 2 Comments

Point 1: If possible, I suggest to add two references in the introduction, both of them talk about COVID and IBD (PMID: 33776370 and 33859104).

Response: We thank the reviewer for the positive review of our paper. Suggested papers have been cited.

Reviewer 3 Report

A very wonderfully summarized review. Please consider adding just one additional piece of information. Currently, there are discussions regarding not only the third additional dose of vaccination against SARS-Cov-2, but also the fourth additional dose. There is probably little information on the efficacy and safety of additional vaccinations limited to IBD patients. However, it would be a good review if you could review as much as possible regarding the third additional vaccination, etc.

Author Response

Response to Reviewer 3 Comments

Point 1: Please consider adding just one additional piece of information. Currently, there are discussions regarding not only the third additional dose of vaccination against SARS-Cov-2, but also the fourth additional dose. There is probably little information on the efficacy and safety of additional vaccinations limited to IBD patients. However, it would be a good review if you could review as much as possible regarding the third additional vaccination, etc.

Response: We thank the reviewer for the positive review of our paper. Available papers on efficacy and safety of additional vaccinations have been discussed and cited.

Reviewer 4 Report

Major:

  • References 3-6 in the introduction are from the early phase of the pandemic. While generally correct, I would advise the authors to cite some of the SECURE-IBD registry publications outlining the association of medication classes with outcomes of COVID. These publications give a more balanced appraisal of risks.
  • With the emergence of booster vaccinations, it is even more important to precisely define what is meant by “fully vaccinated” and what by “booster” (essentially a third dose for all vaccines except for Janssen)
  • Data on COVID vaccinations in IBD patients may be described as limited compared to the general population. Nevertheless, the number of published patients with IBD who received the vaccine vastly outnumbers any of the Phase 3 trials for monoclonal antibodies. A recent meta-analysis (Jena et al. Clin Gastroenterol Hepatol 2022) identified 9447 patients. You may want to cite this publication in your paper.
  • Different assays with diverging reference limits were used in different studies. The correlation between antibody titre and protection from subsequent infection is incompletely characterized for many of the assays. This could be mentioned as a limitation or caveat in the interpretation of findings. You allude to this by citing the Khan & Mahmud study, but perhaps you could formulate this concept more clearly.
  • What about JAK inhibitors and S1P modulators? If data from IBD are sparse, you could support your paper by citing studies from rheumatology and neurology.
  • The vaccination recommendations by BSG/ECCO/IOIBD were made in time of vaccination scarcity which led to the recommendation of taking any vaccine at the first opportunity – the situation is now different and this caveat should be mentioned.

Minor:

  • I suggest adding vaccine type/mechanism of action (mRNA/vector-based/recombinant protein) to Table 1.
  • Are there any data on the association between anti-TNF trough concentrations and subsequent immune response?

Author Response

Response to Reviewer 4 Comments

Point 1: References 3-6 in the introduction are from the early phase of the pandemic. While generally correct, I would advise the authors to cite some of the SECURE-IBD registry publications outlining the association of medication classes with outcomes of COVID. These publications give a more balanced appraisal of risks.

Response: We thank the reviewer for the review of our paper. Paper from the SECURE-IBD registry has been cited.

Point 2: With the emergence of booster vaccinations, it is even more important to precisely define what is meant by “fully vaccinated” and what by “booster” (essentially a third dose for all vaccines except for Janssen).

Response: Precise definition has been included as requested.

Point 3: Data on COVID vaccinations in IBD patients may be described as limited compared to the general population. Nevertheless, the number of published patients with IBD who received the vaccine vastly outnumbers any of the Phase 3 trials for monoclonal antibodies. A recent meta-analysis (Jena et al. Clin Gastroenterol Hepatol 2022) identified 9447 patients. You may want to cite this publication in your paper.

Response: Suggested paper has been cited.

Point 4: Different assays with diverging reference limits were used in different studies. The correlation between antibody titre and protection from subsequent infection is incompletely characterized for many of the assays. This could be mentioned as a limitation or caveat in the interpretation of findings. You allude to this by citing the Khan & Mahmud study, but perhaps you could formulate this concept more clearly.

Response: Suggested limitation has been highlighted as requested.

Point 5: What about JAK inhibitors and S1P modulators? If data from IBD are sparse, you could support your paper by citing studies from rheumatology and neurology.

Response: The proper study concerning response to the COVID-19 vaccine among IBD patients treated with JAK inhibitors has been discussed and cited. Unfortunately, we have not found any studies concerning association between ozanimod used in IBD treatment and COVID-19 vaccines’ safety and efficacy.

Point 6: The vaccination recommendations by BSG/ECCO/IOIBD were made in time of vaccination scarcity which led to the recommendation of taking any vaccine at the first opportunity – the situation is now different and this caveat should be mentioned.

Response: Suggested caveat has been highlighted as requested.

Point 7: I suggest adding vaccine type/mechanism of action (mRNA/vector-based/recombinant protein) to Table 1.

Response: The proper column has been added to Table 1.

Point 8: Are there any data on the association between anti-TNF trough concentrations and subsequent immune response?

Response: The proper paper assessing correlation between serum trough drug levels of anti-TNFs and SARS-CoV-2 antibody concentrations has been discussed and cited.

Round 2

Reviewer 4 Report

The authors have adequately addressed all comments.